



# Leveraging Social Media for Geoscience Communication: Insights from the British Geological Survey's Multi-Hazard and Resilience Campaigns

Eleanor A. Dunn[1], Sam. Illingworth[2], Jon-Paul Orsi[3]

[1]Geophysics Section, School of Cosmic Physics, Dublin Institute for Advanced Studies, Dublin, D02 Y006, Ireland
[2]Department of Learning and Teaching Enhancement, Edinburgh Napier University, Edinburgh, EH11 4BN, Scotland
[3]British Geological Survey, The Lyell Centre, Research Avenue South, Edinburgh, EH14 4AP, Scotland

*Correspondence to*: Eleanor A. Dunn (edunn@cp.dias.ie)



**Abstract** Social media offers a unique avenue for scientific communication; however, it remains underutilised by many scientific organisations. This study examines the social media strategy of the British Geological Survey (BGS), the UK's

leading geoscience organisation, to assess its effectiveness in engaging the public with research on Multi-Hazard and Resilience. We investigate two key research questions; 1. how effectively does BGS engage the public through its social media efforts, and 2. what challenges does BGS face in using these social media platforms to enhance public understanding?

Scientific organisations often rely on the deficit model of communication, characterised by a one-way transfer of knowledge. Yet, emerging studies suggest that a dialogue-based approach, tailored to different social media platforms and formats, may

foster better public engagement. This paper provides a framework for assessing social media activity that can be applied to scientific organisations worldwide.

To address research question 1, we conduct content and sentiment analysis on BGS social media posts – including X, Facebook, LinkedIn, Instagram, YouTube, and BlueSky – from May 2023 to March 2024. A systematic codebook is developed to categorise descriptive and interpretive variables for any social media output. To answer research question 2, we conduct

semi-structured interviews with five BGS employees who manage departmental social media accounts to understand their attitudes towards social media engagement.

Our findings suggest several actionable strategies, such as streamlining communication across platforms, maximising the reach of 'Multi-Hazard and Resilience' themes, increasing video content output, and better incorporating public feedback. Although focused on BGS, our mixed-methods approach and methodology offer a valuable template for other scientific

organisations seeking to enhance their online presence and science communication efforts. This study highlights BGS's successful establishment of a multi-platform online presence, showcasing a range of content formats that effectively engage audiences.

**Short Summary** This study looks at the social media strategy of the British Geological Survey (BGS). We look at how BGS

engage the public through its social media campaigns. We analyse the content in BGS social media posts and conduct interviews with BGS employees to understand their attitudes towards social media. Our results suggest increasing video content output and incorporating public feedback. This research acts as a template for other scientific organisations seeking to enhance their online presence.

## 1 Introduction

The British Geological Survey (BGS) is the UK's premier geoscience organisation and is a partly publicly funded body that aims to advance geoscientific knowledge on the UK landmass using surveying, monitoring, and research (British Geological Survey, 2024). BGS works at local, regional, national, and international scales to tackle several research themes including Environmental Change, Decarbonisation, Multi-hazards and Resilience, and Digital Geoscience. BGS describes its vision as "To be a leading and trusted provider of geological data and knowledge to meet the societal needs for a sustainable future",



achieving this by making its geoscientific research open and accessible to the public. To remain open and public-facing, BGS utilises social media and is active on X (formerly Twitter), Facebook, LinkedIn, Instagram, YouTube, and BlueSky. All BGS accounts are run and maintained by the Communication and Outreach team while departmental accounts are run by individual BGS researchers on a voluntary basis.

This paper offers an independent perspective on BGS' current social media output from May 2023-March 2024, with a focus
on output that falls under the 'Multi-hazards and Resilience' research theme. The lead author is not a BGS employee, so is able to provide an objective overview of social media activity and its effectiveness in reaching the public. To provide this perspective, we will answer two research questions:

**Research Question 1 (RQ1): What patterns of one-way engagement (likes, shares, comment volume) do BGS social**
**media posts receive across different multi-hazard topics?**
**Research Question 2 (RQ2): What challenges does BGS face in using social media to promote their research, and how can these be addressed to improve public engagement?**

Social media is steadily becoming people's primary news source, with its use as a news source increasing yearly (European
Parliament, 2023; Pew Research Centre, 2023; Statista, 2024). BGS, like many scientific organisations, have already incorporated social media into their outreach programmes, with the understanding that a presence on social media is necessary to achieve its vision of providing geological data and knowledge to the public.

To answer the research questions above, this paper uses a mixed-methods approach. Content and sentiment analysis (Hseih and Shannon, 2005; Bryman, 2012) is carried out using social media analysis on posts attributed to the BGS accounts run by
the Communication and Outreach team (hereby known as the 'core accounts') between May 2023-March 2024. Social media analysis aims to draw conclusions on what Multi-hazard and Resilience themes the public engages with most and to understand the attitudes associated with different themes. Semi-structured interviews are used to understand the attitude of BGS employees towards using social media as part of their research ethos. These employees self-identify as the most involved in running the departmental social media accounts ('non-core accounts'). The interviews are used to understand what barriers individuals
may face when using social media to promote their research and how these barriers can be tackled.

Social media platforms offer scientific organisations unprecedented reach to disseminate findings and gauge public interest through measurable interactions such as Likes, Shares, and Comments. While true conversational exchange requires dedicated reply strategies, many agencies start by assessing one-way engagement metrics to understand which topics resonate most with audiences. In this study, we analyse BGS's Twitter and Facebook accounts to investigate 1) how engagement levels differ
across hazard topics, and 2) the sentiment expressed by the public in comment threads.





This paper is designed to act as a framework that can be applied to national and international scientific organisations to allow them to understand and assess their social media activity going forwards.

## 2 Social Media and Science Communication

Social media for scientific communication, is a relatively recent science communication strategy, however, some critics
suggest that it is still not used to its full potential (Lee and VanDyke, 2015; Brown Jarreau et al., 2019, Höttecke and Allchin, 2020; Márquez and Porras, 2020). For successful scientific engagement with the public, the dialogue model is preferred over the deficit and participation models. The deficit model implies a one-way communication between the scientists and the public (Durant et at., 1992; Millar and Wynne, 1998) and assumes that the public has a lack of knowledge and needs to be supplied with the science via dissemination (Callon, 1999; Bucchi, 2004; Irwin, 2008; Joly and Kaufmann, 2008; Trench, 2008). The
dialogue model creates an equal playing field between scientists and the public and involves the sharing and co-production of knowledge (Joly and Kaufmann, 2008; Bubela et al., 2009). The participation model follows the dialogue model but involves enabling participatory mechanisms for deliberating on science (Bucchi and Trench, 2016). More recent studies acknowledge that there may not be a clear distinction between these three models and then a spectrum of outreach from dissemination to participation may be more applicable (Metcalfe, 2019; Illingworth, 2023).


Although two-way dialogue on social media can enhance public trust and co-create knowledge, it entails labour-intensive response protocols that exceed the scope of this initial study. Instead, we focus here on quantifiable indicators of audience engagement – namely post reactions (Likes/Shares), Comments, and Comment sentiment – as a first step toward understanding how BGS content performs across platforms and hazard topics.

### 2.1 Geoscience Communication

The British Geological Survey is a geoscience organisation and hence any of its communication research becomes geoscience communication. With growing concerns about our environment comes an increasing number of scientists who want to facilitate a valuable exchange of knowledge between geoscientists and non-geoscientists. The journal 'Geoscience Communication' was launched in April 2018 to tackle this issue. Since its launch, the journal has published papers which delve into the different
forms of science communication, as described above (Illingworth et al., 2018).

### 2.2 Social Media and Scientific Organisations

Many governmental organisations still use the deficit model when using social media and continue to practise one-way communication, underutilising social media's potential. Several studies have focussed on the use of social media for science communication by U.S. federal Governmental scientific agencies and found that agencies did not engage with one-on-one
dialogue; based on a limited number of replies to comments and questions left by public users, instead treating the platforms as broadcast media (Lee and VanDyke, 2015; Shin et al., 2015; Lee et al., 2017).



Various studies have examined the potential for a dialogue-based approach to scientific engagement on social media but these approaches will depend on the type of social media being used and how the same scientific information can be repackaged into different formats to engage with various audiences (Höttecke and Allchin, 2020; Márquez and Porras, 2020; Zawacki et al., 2022).

## 2.3 Social Media and BGS

BGS are currently active on X, Facebook, LinkedIn, Instagram, YouTube, and BlueSky. Below we provide a background to the different social media platforms and provide examples of how they have been successfully used to practise science communication.

### 2.3.1 X (Formerly Twitter)

X was established in April 2007, and by 2023, boasted approximately 556 million monthly active users, with 58% of the user-base under 35 years old (Dixon, 2024). In October 2022, X was bought by Elon Musk for $44 billion, resulting in a 23% decrease in monthly users within the first year of Musk's takeover (Shah, 2023). X is a short-form text-based platform, allowing users to post 280 characters or less, each post can include a maximum of four pictures and/or one video, capped at 140 seconds in length. In October 2023, Musk introduced X Premium, which for a paid membership, allowed the user to post a maximum of 25,000 characters, experience reduced ads, and have access to X's AI feature. The primary search mechanism on X involves hashtags (#), users can discover trending hashtags through the 'Explore' tab.

Su et al. (2017) analysed scientific institutions' use of X for one-way and two-way communication in connection to science festivals from 2012-2015. The deficit model was the primary model used where many posts on X would be used to distribute information and promote festival events. In comparison, Watson et al., 2023 studied the use of X for disseminating Interferometric Synthetic Aperture Radar (InSAR) data to non-expert users. This study was able to identify who was interacting with disseminated data, how the data was being used, and to propose strategies for meaningful communication. By focussing on a niche science, Watson et al. (2023) were able to manage their data capabilities and make meaningful conclusions specific to a select group of scientists while Su et al. (2017) used a broad topic to make overarching conclusions about how methodologies may need to be changed going forwards.

### 2.3.2 Facebook

Facebook was founded in January 2004. By 2023, it had approximately 3,049 million active monthly users, with 31% in the 25 – 34 age category (Dixon, 2024). As one of the oldest social media networks that is still incredibly popular, Facebook has undergone several transformations since its inception, yet its emphasis on text and picture posts remain significant. As well as posts to a user's 'feed' (which remain permanent unless deleted) users can post temporary 24-hour stories. Facebook has made



a recent shift to prioritising 'Facebook video' to compete with other popular video-based platforms such as TikTok (Facebook, 2015). Hashtags are used on Facebook but the primary focus on the platform is one's individual feed and the activity of 'Friends'. Facebook's parent company Meta, also owns Instagram, allowing users to connect their accounts across platforms and cross-post.

Facebook is the largest social media network worldwide and is still the most popular social media platform in many countries, allowing scientists to conduct global studies (Vu et al., 2020; Khosla and Pillay, 2020; Connoway et al., 2022; Graham et al., 2024). Vu et al. (2020) analysed 289 global Non-Governmental Organisations (NGOs) and their framing of climate change. With the use of content analysis, the authors can conclude that climate impacts are more likely to appear in NGO's persuasive messaging that efficacy.

Unfortunately, Facebook is also leading the way in dissemination of fake news and pseudoscience (Khosla and Pillay, 2020; Connoway et al., 2022; Rauchfleisch et al., 2023) and caution should be taken to tackle fake news and misinformation.

### 2.3.3 Instagram

Instagram, founded in October 2010, currently has approximately 2,000 million active monthly users, with 31.8% falling in the 18 – 24 age group (Dixon, 2024). In September 2012, Facebook acquired Instagram for $300 million and 23 million shares (Protalinksi, 2012). Instagram is a visual-based platform that allows users to upload 1 – 20 pictures, alongside a text caption. Like Facebook, users can post a temporary 24-hour 'story', accessible by clicking on the user's profile picture. In April 2020, Instagram introduced 'Instagram Reels', enabling users to upload up to 90 seconds of content set to audio. This feature was introduced to complete with platforms such as TikTok. The 'Search' tab on Instagram is used to find users and popular hashtags.

Brown Jarreau et al. (2019) identifies Instagram's potential for public engagement by highlighting how science museums are missing an opportunity to promote education, scientific literacy, public engagement, and the public visibility of museum scientists. The authors suggest that museum's utilise Instagram's features, including 'Stories' and 'Reels', to highlight museum researchers and delve into storytelling of scientific discovery in action, with public input.

Another observation by Caspari (2022) is that defining a target audience and developing a niche is essential in a social media communication strategy. However, this may be difficult for scientific organisations, such as BGS, that want to promote the broad scope of their research. Caspari (2022) also found that regular activity is a critical factor in maintaining engagement, with accounts that only post once every week/month unlikely to reach a broad audience and grow their follower base. Institutions also receive lower engagement than individual accounts, emphasising the need for organisations to work even harder to generate Instagram engagement.





### 2.3.4 YouTube

Established in February 2005, YouTube is a video-based social media platform where approximately 2,500 new videos are uploaded to the platform every minute (Hayes, 2024). As of 2023, it boasts roughly 2,491 million active monthly users (Dixon, 2024) with 54.3% of users falling in the age range of 18 – 34 (Dean, 2024). With a primary focus on video creation and consumption, YouTube stands out from the other social media platforms used by BGS. Users are presented with a homepage featuring recommended videos, and they can utilise the 'Creator Studio' to record, edit, and upload their video content. Users

can 'Subscribe' to channels they wish to follow and receive notifications when new content is uploaded from those channels.

There is a plethora of studies which look at popular science channels and why certain YouTube videos/channels receive the highest level of engagement (Newman and Schwarz, 2018; Beautemps and Bresges, 2021; Pattier, 2021; Fischer et al., 2024). YouTube allows for an added level of creativity, which may not be possible with text or image-based posts, with Allgaier (2013) writing about the popularity of science in music videos, which can be easily shared and are typically short while still

containing a wealth of information.

### 2.3.5 BlueSky

BlueSky is the most recent social media platform that BGS has signed up to. It was established in 2021, in respond to Musk's takeover of X. BlueSky features a similar interface to X, allowing users to access similar features and content, even giving the

user the opportunity to search for any followers they previously had on X that may have moved over to BlueSky. Prior to February 2024, it was only possible to set up a BlueSky account using an invite link from an existing user – this was during its 'Beta' period – but the platform is now open to everyone. BlueSky currently boasts 5.2 million users, with the most significant demographic falling within the 18 – 24 age group (Ver Meer, 2024). However, the number of users continues to grow as many X users migrate to BlueSky.

BGS has a substantial online presence on multiple social media platforms and this paper will look at how each platform is being used and which research themes receive the most positive engagement. By combining content and sentiment analysis with semi-structured interviews, a picture can be built on the current stage of BGS' online presence and how this may change going forwards.

### 3 Methodology

### 3.1 Social Media Analysis

To answer RQ1, we perform content and sentiment analysis on posts attributed to the BGS' social media accounts (X, Facebook, LinkedIn, Instagram, YouTube, BlueSky) between May 2023-March 2024. Table 1 provides an overview of the





accounts run and maintained by the BGS Communication and Outreach team as well as accounts run by individual BGS

departments.

| Platform | Account Handle | Followers | Following | Posts | Active Since |
|---|---|---|---|---|---|
| X | @BritGeoSurvey | 51,400 | 1,172 | 23,400 | February 2009 |
| Instagram | @britgeosurvey | 8,045 | 182 | 329 | June 2014 |
| Facebook | British Geological Survey | 33,000 | 31,000 | - | July 2009 |
| LinkedIn | British Geological Survey | 41,901 | - | - | - |
| YouTube | British Geological Survey | 8,970 | - | 121 [Videos] | August 2008 |
| BlueSky | @britgeosurvey.bsky.social | 428 | 193 | 34 | October 2023 |
| Not run by BGS Comms | | | | | |
| X | @BGSVolcanology | 7,330 | 349 | 1,044 | March 2013 |
| X | @BGSseismology | 1,601 | 99 | 599 | July 2016 |
| X | @BGSspaceweather | 6,732 | 126 | 3,589 | December 2010 |
| BlueSky | @bgsspaceweather.bsky.social | 3 | 1 | 0 | February 2024 |
| X | @BGSAuroraAlert | 11,400 | 92 | 352 | December 2010 |
| BlueSky | @bgsauroraalert.bsky.social | 1 | 1 | 0 | February 2024 |
| X | @BGSLandslides | 2,573 | 832 | 4,657 | July 2012 |
| BlueSky | @bgslandslides.bsky.social | 21 | 31 | 3 | February 2024 |
| X | @BGSMarineGeo | 1,936 | 1,059 | 4,162 | August 2024 |

**Table 1 An overview of all social media accounts associated with the British Geological Survey. Research departments that run their own accounts are Volcanology, Seismology, Geomagnetism, Landslides, and Marine. Followers, Following, and Posts are all recorded on the 27th of March 2024.**



We use systematic content analysis as described by Bryman (2012) by creating a codebook of descriptive and interpretive variables. The codebook variables are listed in Table 2. Content analysis is regarded as a flexible and practical research tool to interpret textual data (Hseih and Shannon, 2005) and has been used effectively in previous studies exploring online vaccine sentiment (Wolfe et al., 2002; Hoffman et al., 2019; Smith and Graham, 2019) and environmental activism (Vu et al., 2020). The codebook is based on specific research themes highlighted on the BGS website – Decarbonisation and Resource

Management, Environmental Change, Digital Geoscience, National Geoscience, International Geoscience, and Multi-Hazards and Resilience. National Geoscience and International Geoscience are not included as individual content themes within the codebook because they were deemed too broad and overarching. These research themes are not independent of each other and social media posts may be considered to represent multiple research areas. This study is primarily focussed on 'Multi-Hazards and Resilience', therefore Figure 1 demonstrates how research areas within Multi-Hazards and Resilience (Earthquakes and

Seismology, Geomagnetism, Volcanology, Landslides, Geodesy and Earth Observation, Shallow Geophysics, Shallow Geohazards, Coasts and Estuaries) intersect with other BGS themes. Figure 1 also provides a clear distinction between research areas which may be seen as very similar in topic e.g., Shallow Geophysics vs. Shallow Geohazards.

The lead author analysed BGS social media posts that were published between May 2023 – March 2024 using the codebook outlined in Table 2. 50 random social media posts are selected and analysed at the beginning of the data collection period and

then analysed a second time, four weeks later, to ensure correct content and sentiment identification, before the remaining posts were categorised. Each post is given a Unique Item Identifier; their URL and descriptive variables such as the number of likes, views, comments, shares, and video run time are recorded. For Facebook and LinkedIn, the user is given various 'React' options for a post (compared to other platforms where the only option is to 'Like'); in this case, a 'Like' is equal to all positive 'Reacts' that are available to the user e.g., Love, Care, Haha. Other descriptive variables including any attachments

to the social media post (pictures, weblinks etc.) and the inclusion of hashtags are not incorporated into the final analysis.

To identify the correct research theme for content analysis, the social media posts are viewed at face value – as in it is assumed the viewer does not click on any attached information to understand the posts' theme further. A short post on X, with a maximum of 240 characters may not provide the level of detail provided on Facebook, LinkedIn, or Instagram therefore the research theme may be different across platforms. For YouTube videos, the video is watched in its entirety to establish its

content. Some social media posts do not fall into any research discipline so are not included in the results section, e.g., a social media post notifying the public about Christmas opening times.

For sentiment analysis, the comments/replies to each post are analysed. The comment categories are 'Positive', 'Negative', 'Neutral' and 'Spam'. 'Positive' comments may be denoted by a positive emoji e.g., smiley face, laughing face or by showing interest in the social media post content. 'Negative' comments often involve disagreeing with the content, such as climate

change denial or frustration with the BGS data portal not working. 'Neutral' comments are either associated with 'tagging' a friend (this will give the friend a notification to view the post) or asking a question,  in this case it cannot be ascertained whether the 'tagging' or question agrees or disagrees with the original post. 'Spam' comments are defined as replies that spread



"bulk unsolicited contents usually for the purpose of advertisements, promoting pornographic websites, fake weight loss, bogus donations, fake news, online job scams, and a host of other malicious intents, which are perpetrated by spammers." (Adewole et al., 2017) Spam messaging can be spread by real people and bots (automated accounts). These comments can be established as belonging to 'bots' by clicking on the profile and confirming the accounts validity. Since Musk's takeover of X, a rise in 'bot' accounts has been observed (Hickey et al., 2023).

|  | Code | Applicable on which platform? |
|---|---|---|
| **1** | Unique Item Identifier [URL] | All Platforms |
| **2** | Date of publication [dd/mm/yyyy] | All platforms |
| **3** | Number of likes | All platforms |
| **4** | Number of views | YouTube |
| **5** | Number of comments | All platforms |
| **6** | Number of shares | X, Facebook, LinkedIn, BlueSky |
| **7** | Video Run Time | YouTube |
| **8** | Is the post the start of a thread? | X |
| **9** | Is a picture/video attached? | X, Facebook, LinkedIn, BlueSky |
| **10** | Does the post contain any #? | All platforms |
| **11** | Is the post related to a BGS research theme? | All platforms |
| **12** | If yes to 11, what discipline? | All platforms |
| **13** | If answer to 12 is 'Multi-hazards and resilience' then which multi-hazard theme? | All platforms |
| **14** | Does the caption/visual present substantive scientific information, science factors or scientifically educational material? | All platforms |
| **15** | Number of positive comments | All platforms |
| **16** | Number of negative comments | All platforms |
| **17** | Number of neutral comments | All platforms |
| **18** | Number of spam comments | All platforms |

**Table 2 The codebook used to record variables for each social media post. Some descriptive variables differ per social media platform due to the nature of the platform.**




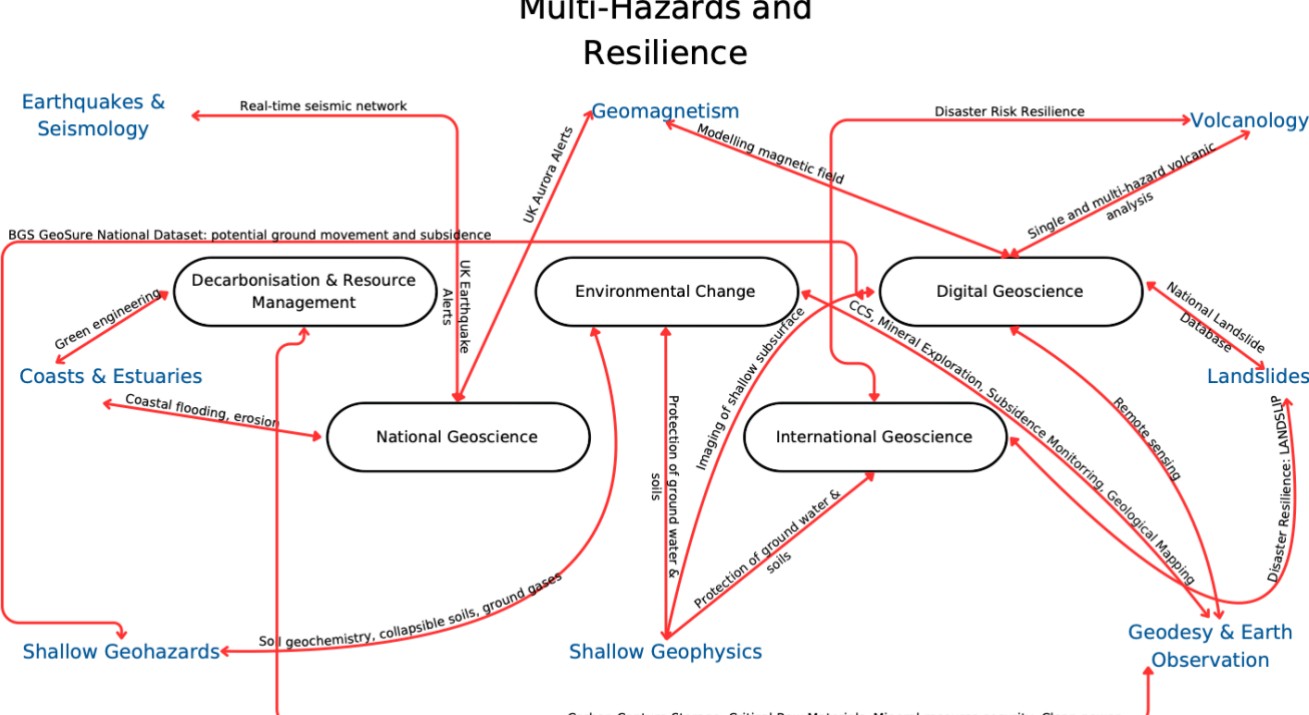

**Figure 1 The different research themes that fall under the British Geological Survey Research umbrella. A red arrow indicates how an aspect of 'Multi-Hazards and Resilience' (displayed on the boundaries of the figure) may connect to another theme e.g., the National Landslide Database is run by the Landslide Research Group and is part of the Digital Geoscience group.**


### 3.2 Semi-Structured Interviews

To answer RQ2 and understand a small selection of the challenges that scientific organisations may face when undertaking science communication, this study uses semi-structured interviews. Five BGS employees took part in a semi-structured interview to understand current attitudes towards corporate social media use. These employees self-identified as the most

heavily involved in maintaining the departmental BGS social media accounts, as identified in Table 1. Table 3 lists the questions asked during each interview, however, there is no strict interview structure. Each non-core account is hosted on X; hence, the questions are geared towards experiences on that platform. However, some research groups had recently set up accounts on BlueSky at the time of interview. In some instances, research groups split the responsibility for account maintenance, but in other cases, the responsibility fell to one person. Each interview lasted approximately 20 – 30 minutes.

Informed consent was obtained by all interviewees and the questions were outlined to them prior to the interview. The interviews were recorded only for data retention purposes and any audio recordings were destroyed following transcription. It was made clear that the interviewees were free to leave at any point during the interview process. Ethical approval was obtained by the British Geological Survey.





|   | Questions |
|---|---|
| 1 | Who is involved in running the account? Just you or multiple people? |
| 2 | How often do people log onto the account? |
| 3 | Amongst those that use X, is there a consensus of like or dislike of the platform? |
| 4 | Would you/the research group be interested in setting up an alternative social media presence to X such as BlueSky or Instagram? |
| 5 | What sort of things do you post about on the social media account? |
| 6 | Do you/the research group actively engage with the X community? This may include replying to comments, joining in discussions etc. |
| 7 | Would you/the research group be more inclined to utilise the platform if it was part of your job description? |
| 8 | If managing social media was part of your job description how would you improve the research group's social media engagement? |

**Table 3 Questions asked in the semi-structured interviews.**

## 4 Results

### 4.1.1 Social Media Analysis

Figure 2 displays the Likes/Positive Reacts and Comments received over time across all social media platforms utilised by the
BGS to provide an overview of activity on each platform between May 2023-March 2024. Figure 2 also provides an overview
of the platforms that are used frequently (X, Facebook, LinkedIn, BlueSky) compared to the platforms used infrequently
(Instagram, YouTube). Despite infrequent activity, some of the highest numbers of Likes are attributed to Instagram. LinkedIn
also appears to receive some of the highest numbers of Likes and Comments, with a post regarding availability of 10K maps
on the BGS map portal receiving 1,009 Likes and 28 Comments. Another post which received a high level of engagement, in
comparison to the average level of engagement received on BGS accounts, was on Facebook, regarding an amateur geologist
recreating a geological map of Scotland using rock samples, which received 999 Likes and 29 Comments. The average number
of Likes and Comments received per platform are 20.9 and 0.6 (X), 39.8 and 1.6 (Facebook), 63.9 and 1.7 (LinkedIn), 4.7 and
0.2 (BlueSky), 219.6 and 3.9 (Instagram), and 6.3 and 0 (YouTube).







**Figure 2 The engagement per platform throughout the period of data collection. 2.A Number of Likes against time per platform. 2.B Number of Comments against time per platform.**



Figure 3 looks at the number of social media posts attributed to each research theme (Decarbonisation, Digital Geoscience, Environmental Change, Multi-Hazards and Resilience) across platforms. On each platform, Multi-Hazards makes up the highest number of posts, with Decarbonisation making up the second highest posts across X, LinkedIn, and Facebook.


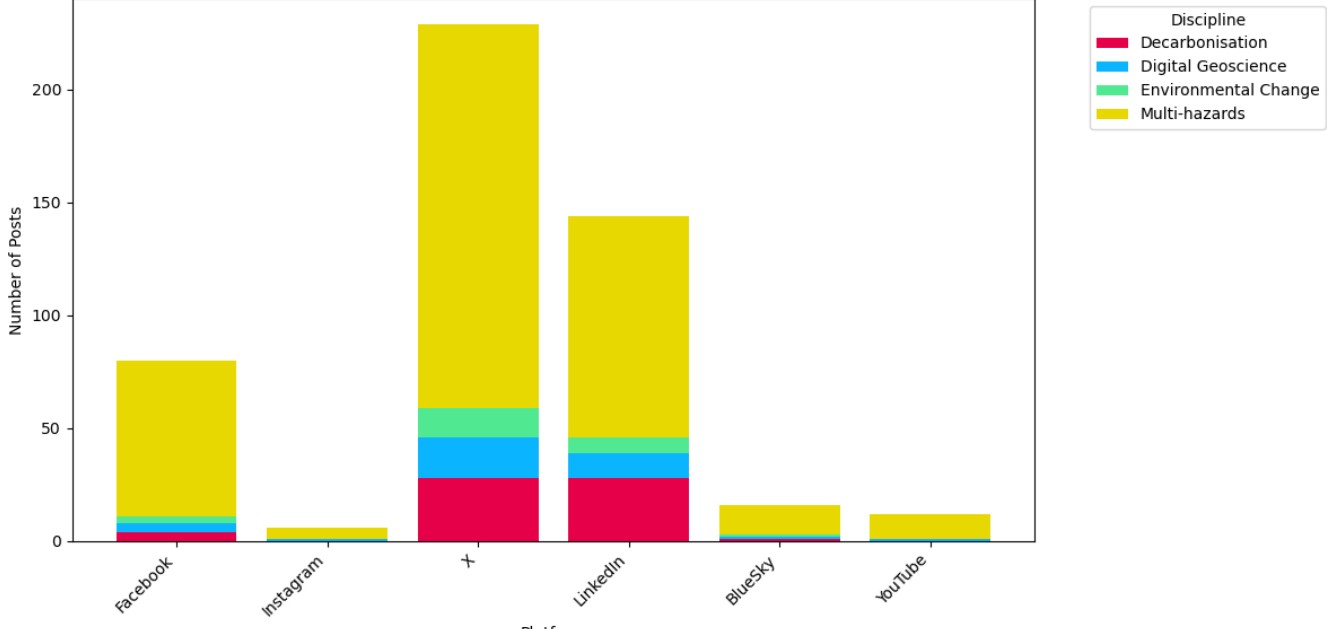

**Figure 3 The research discipline associated with each social media post across all platforms.**

When examining the spread of research areas within the Multi-Hazards and Resilience discipline (Earthquakes and Seismology, Geomagnetism, Volcanology, Landslides, Geodesy and Earth Observation, Shallow Geophysics, Shallow Geohazards, Coasts and Estuaries) the bias can be observed in Figure 4. Geodesy and Earth Observation make up most social media posts across all platforms; this is in large part due to the broad scope of the research area which according to the BGS website, incorporates mine-waste mapping, subsidence monitoring, shale gas impacts, mineral exploration, $CO_2$ storage, mineral mapping, geological mapping, and remote sensing. Earthquakes and Seismology is a popular area on text-based platforms, making up the second highest % on X, BlueSky, and Facebook. Announcements of local seismic events are most suitable for X's short character limit and would not perform well on image-based platforms, unless the information was transformed into a graphic. However, seismic activity is typically time sensitive so releasing this information in a timely manner would be the best option. Volcanology makes up 1/5 of all Instagram posts, perhaps due to the field's visually appealing nature, making the research area easily accessible on image-based platforms.

To analyse the Likes per Post based on the Multi-Hazard and Resilience Theme (Figure 5), the Likes are normalised to account for the differing number of posts per platform. Figure 5 looks at the Multi-Hazard content distribution across the different








social media platforms utilised by BGS. Surprisingly, Landslides receives the highest proportion of Likes despite a very small number of posts (Figure 4). Similarly, Volcanology proved popular on Facebook, Instagram, X, and LinkedIn. However, Geodesy and Earth Observation, while contributing to a large proportion of total posts on each social media platform, does not receive a similarly high level of engagement.

Figure 6 examines the Comment sentiment on all Multi-Hazard and Resilience social media posts. The number of Comments received per theme is split into 'Positive', 'Negative', 'Neutral', and 'Spam', as described in the methodology. All themes, except Coasts and Estuaries and Shallow Geophysics, receive a majority of 'Positive' comment sentiment. The highest number of 'Negative' comments are attributed to Coasts and Estuaries (50%) and the highest number of 'Spam' comments was attributed to Earthquakes and Seismology (15.3%).


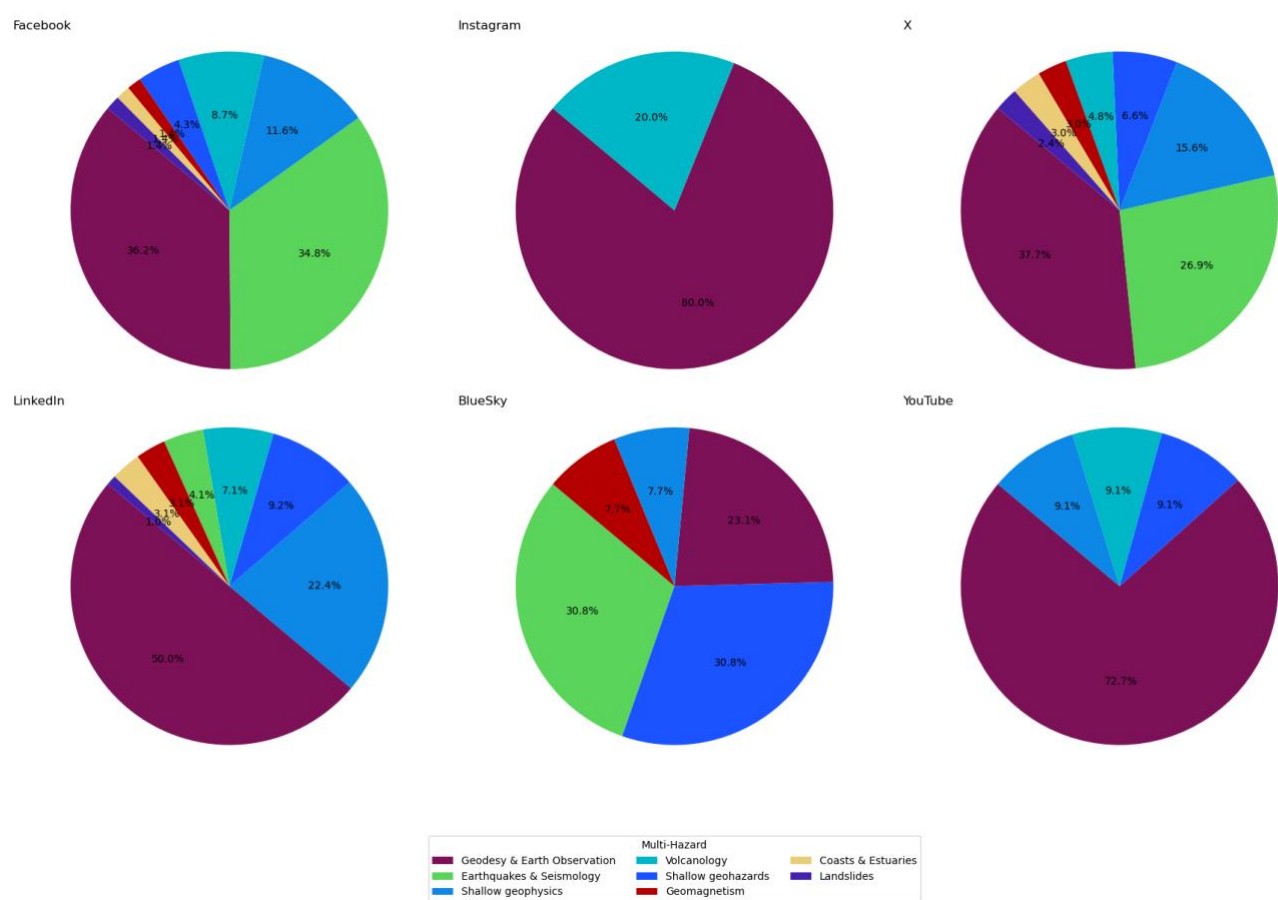

**Figure 4 Pie charts for each social media platform. Each pie displays the distribution (%) of social media posts attributed to each research discipline under the 'Multi-Hazards and Resilience' research umbrella.**



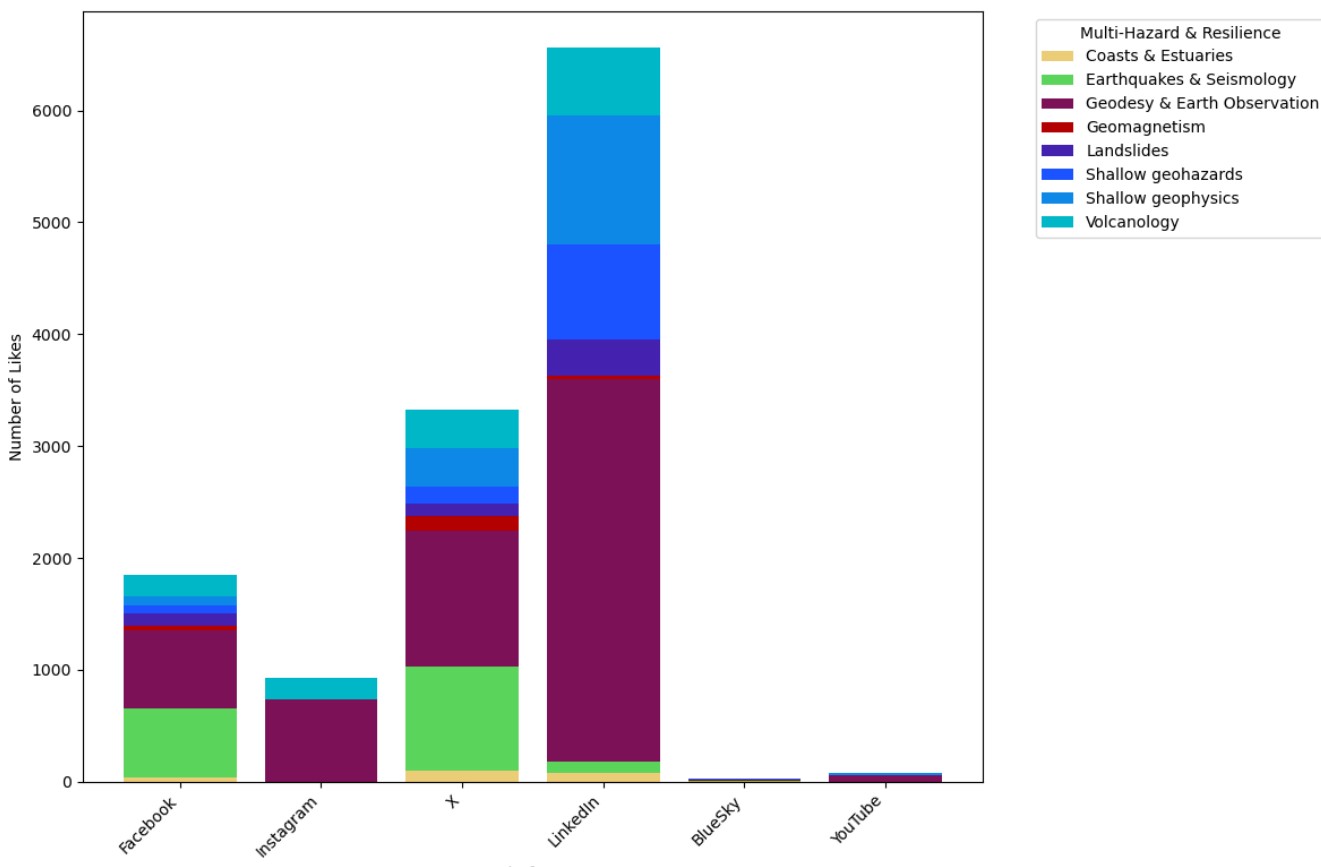

**Figure 5 The number of Likes attributed to each Multi-Hazard and Resilience research theme per platform.**



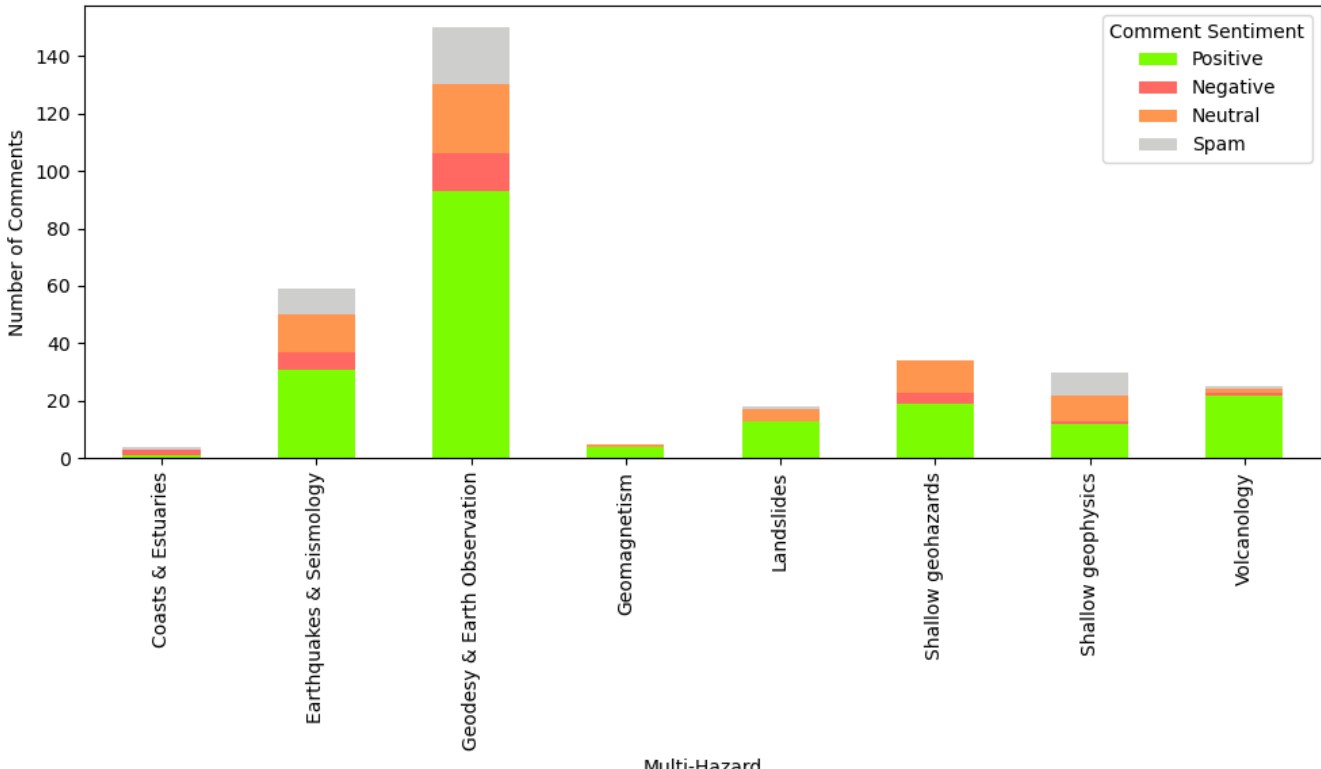

**Figure 6 The average number of Comments (normalised by total number of social media posts) for each Multi-Hazard and Resilience theme, split into Positive, Negative, Neutral, and Spam Comment sentiment.**

However, this could be attributed to a large portion of Earthquakes and Seismology related posts being published on X, which
has experienced an uptick in spam activity.

### 4.1.2 Social Media Analysis

The Chi-squared Goodness-of-fit is calculated for the results presented in Figures 5 and 6. For Figure 5, a Chi-squared value of 1199.57 informs us that there is a big difference between what we observed (the Likes attributed to each platform) and what
we expected (a uniform distribution where Likes are spread equally across all platforms). The p-value is $3.643 \times 10^{-257}$, meaning that the difference observed is unlikely due to random chance – creating an extremely significant result. This suggests that some platforms are receiving significantly less positive engagement than others, as shown in Figure 5. This leads to the first suggestion that an increase in posts is implemented across all platforms, perhaps streamlining the content best suited for visual (Instagram), short-form text (X, BlueSky), long-form text (Facebook, LinkedIn) and video (YouTube). Table 4 shows the Chi-
squared and p-values associated with the Positive, Negative, Neutral, and Spam Comment Sentiment, as detailed in Figure 6.





| Comment Sentiment | Chi-squared Statistic | p-value |
|---|---|---|
| **Positive** | 247.46 | $9.61 \times 10^{-50}$ |
| **Negative** | 40.26 | $1.12 \times 10^{-6}$ |
| **Neutral** | 59.54 | $1.87 \times 10^{-10}$ |
| **Spam** | 69.6 | $1.78 \times 10^{-12}$ |

**Table 4 The Chi-squared statistic and p-values associated with the distribution of Positive, Negative, Neutral, and Spam Comment sentiment, as detailed in Figure 6.**

In Table 4, all p-values are must smaller than 0.5, suggesting that for every Comment sentiment, the distribution of Comments across Multi-Hazard and Resilience themes is significantly unequal, where some research themes receive a lot more comment than others. The Chi-squared Statistic implies that this is not due to random chance.

These results indicate that different Multi-Hazard and Resilience themes generate very different responses from the audience. In some instances, certain Multi-Hazard and Resilience themes e.g., Landslides and Volcanology may trigger more Positive sentiment. In contrast, others may attract more Negative or Neutral comments e.g., Shallow Geohazards and Shallow Geophysics, while other themes may be prone to Spam e.g., Coasts and Estuaries. Table 5 lists some examples of Positive comments on Landslides, Negative comments on Coasts and Estuaries, Neutral comments on Shallow Geohazards, and Spam comments on Shallow Geophysics.

| Multi-Hazard Theme | Comment Sentiment | Comment Example | Platform |
|---|---|---|---|
| **Landslides** | | This is great to see 👍 | LinkedIn |
| **Geodesy and Earth Observation** | Positive | Sir please be my guest when ever you plan to visit Himalaya subduction zone. I am from kullu valley, India.. And rock hounder and insetu gem digger | YouTube |
| **Volcanology** | | What a job 😄 | Instagram |
| **Coasts and Estuaries** | Negative | I hope there are some stiles to give beach access.... | X |
| **Earthquakes and Seismology** | | No postal order through the post then. | X |





| Shallow geohazards | Neutral | [Tagging Friend] | Facebook |
|---|---|---|---|
| Geodesy and Earth Observation | | [Tagging Friend]? | Facebook |
| Shallow Geohazards | | Presumably this is based on rock type and not actual acquired readings ? | LinkedIn |
| Shallow geophysics | Spam | [Reply hidden due to NSFW content] | X |
| Geodesy and Earth Observation | | Please check your DMs for business regarding your video. Thanks! | Instagram |

**Table 5 Examples of Positive, Negative, Neutral, and Spam Comments left on social media posts attributed to different Multi-Hazard and Resilience research themes. Comments are written as they were on the original social media post, to include all spelling errors and grammatical mistakes.**


## 4.2 Semi-Structured Interviews

Across interviews, multiple themes emerged when asked about the experience of running departmental accounts. I will discuss these themes below.

**4.2.1 Multi-Hazard and Resilience Accounts are Unique**

Each department uses its social media account for different purposes – this would make consolidating them under the single @britgeosurvey account challenging. For instance, @BGSVolcanology shares research activities and engages with volcano observatories, @BGSAuroraAlert provides daily space weather forecasting, @BGSLandslides collects online reports of landslide activity for the daily UK Landslide hazard assessment, and @BGSseismology updates the public on recent UK and

international seismic activity. As each account has different goals, incorporating each account into the main BGS account would overwhelm social media output and become a net negative.





### 4.2.2 A Change from Twitter to X

All interviewees discussed a decrease in account enjoyment following Musk's takeover in 2021. Interviewees mentioned an increase in irrelevant and extremist content on the account timeline and in replies to social media posts. This increase cannot be verified for each account but a general rise in extremist activity has been reported in studies (Graham and FitzGerald 2023; Hickey et al., 2023). It was discussed if a change to a similar platform such as BlueSky may increase social media enjoyment however several reasons were given against switching social media platforms:

- The time required to move to an entirely new (albeit, similar) and unfamiliar platform and build up an audience.
- Not enough of the X community has moved to alternative platforms such as BlueSky.
- Other social media platforms, such as Instagram, are not suitable for the type of content being produced for X.

### 4.2.3 Account Maintenance – Individual or Group Responsibility?

Running and maintaining a successful social media account warrants a certain level of dedication and interest, which, in some instances, may not be widely shared within a given research department. Various factors given for this lack of interest included:

- Lack of understanding/interest in social media.
- A preference for dedicating all work time to research activities rather than social media activity.
- Apprehension about engaging with the public.

To tackle these issues, it is suggested that support be provided in cases where social media maintenance falls on an individual rather than a collective, this will enable the continued success of the account. Additionally, training should be offered to encourage active participation in Multi-hazard and Resilience communication on social media platforms.

### 4.2.4 Dedication to Account Maintenance

A large amount of time is required to run an active social media account on X that engages with the community and posts consistently. Aside from @BGSLandslides, where generating a daily UK Landslide Hazard Assessment is part of the team's remit, all other departmental accounts are run voluntarily. However, when discussing how the account many be improved if its maintenance was a paid opportunity, potential avenues which were mentioned were:

- Dedicating monthly posts to highlighting a team member and their research.
- Shutting down the social media account completely and work towards narrowing the lines of communication to the public.
- Setting up alternative social media accounts e.g., Facebook, Instagram, and BlueSky.

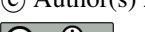



There is a clear interest in increasing communication activities within each department, but this is not a priority when accounts are maintained on a voluntary basis.

## 5 Discussion

### 5.1 Suggestions

The following suggestions combine results from the social media analysis and the semi-structured interviews.

### 5.1.1 Lines of Communication

There is an interest in streamlining lines of communication across departmental social media platforms. Based on social media analysis (figure 4), there is an uneven distribution of posts relating to each Multi-Hazard and Resilience theme. It is suggested that research teams generate their own content that can be incorporated into the main lines of communication. One solution may be to include each Multi-Hazard and Resilience team member on a weekly rota where one person from each department is the 'Communication Lead' for each week. The 'Communication Lead' would require being on call from the 'Communication and Outreach' team to answer any questions received on social media platforms or to offer advice on social media captions/campaigns. Several departments have already implemented an 'on-call' duty so it is suggested that social media responsibility also be incorporated into this roll.

### 5.1.2 Utilising All Platforms

Instagram received a very high number of likes (figure 2), despite inconsistent posting. It is suggested that BGS utilise all of Instagram's features to increase engagement such as Instagram 'Reels' and 'Stories' (Brown Jarreau et al., 2019) as our results suggest that there is an underutilised market that BGS is not tapping into. Caspari (2022) notes that consistent activity on Instagram is critical in maintaining high levels of public engagement. BGS already receives a high level of engagement on Instagram so this suggests that Instagram could become one of BGS' most popular social media platforms if it is used to its full potential. Another platform not being utilised fully is YouTube, which currently receives a very low number of Likes and Comments regarding other platforms. Future study is suggested to understand why this may be. This can be done by analysing BGS YouTube content and looking at average watch time as well as viewership information. BlueSky, although in its infancy as a platform, has the potential to become a contender with X so it should not be neglected and consisting cross-posting with X is encouraged.

### 5.1.3 Distribution of Multi-Hazard and Resilience Themes



According to Figure 4, there is not an even distribution of the different 'Multi-Hazard and Resilience' themes being posted across different social media platforms, but even when accounting for the number of posts (Figure 5), there are discrepancies in the number of Likes each 'Multi-Hazard and Resilience' theme receives. In this case it is suggested that work be done to improve engagement across themes, for example, creating an increase in video outputs.

The Comment Sentiments do tell us that most Comments received are Positive, excluding Coasts and Estuaries. However, this cannot be attributed to dislike of the actual theme as some social media post may warrant Negative sentiment e.g., a social media post relating to a disaster.

### 5.1.4 Video Output

The intricacies of video communication were not explored in this study; however, based on Figures 2 and 4, we believe it is being underutilised and not being used to its full potential as a vital means of communication. It is important to note that producing videos is a lot more time consuming compared to creating written/image-based content. An example of an ongoing video campaign by the British Geological Survey (BGS) is the 'My Career in 60 Seconds', where employees describe their roles within the organisation and their origin story which led them into the career. A suggestion to optimise video content creation is to dedicate one week every quarter to producing and filming video content. These videos can then be released gradually over the following three months, providing a consistent stream of video content without overwhelming resources or creating content gaps. Channel consistency, structure, quality, and present are all crucial to enhancing engagement on YouTube (Newman and Schwarz, 2018; Velho et al., 2020; Beautemps and Bresges, 2021; Pattier, 2021). This can be achieved by uploading video content at a steady rate rather than releasing multiple videos within a short time frame, followed by extended periods of inactivity. Consistent uploads maintains audience interest and encourages regular interaction with the channel.

### 5.1.4.1 Short-Form Video Content

In recent years, platforms like TikTok have sparked a huge surge in short-form video content, prompting Instagram, YouTube, and Facebook to introduce features such as 'Instagram Reels', 'YouTube Shorts', and 'Facebook Video' to encourage uploading of short-form video content. Short-form videos challenge users to convey a narrative within a short timeframe, yet this format has proved highly engaging for geoscience communication (Zawacki et al., 2022). It was recommended that BGS venture into the short-form video content space, these videos may require less time to produce compared to longer-form content. It is also worth noting that while TikTok may not be an option for BGS, most other social media platforms offer similar opportunities for uploading short-form video content. Expanding into this space can broaden BGS's reach and engagement with audiences who prefer consuming content in shorter formats. The tone of the short-form video content should also be considered, and an agreement should be reached between the Multi-Hazard and Resilience teams on communication strategy.




### 5.1.4.2 Multi-Hazard and Resilience FAQs

One idea mentioned during the semi-structured interviews for a future video campaign was Multi-Hazard FAQs. These 5 –
10-minute videos where a representative from a Multi-Hazard and Resilience department answers a series of frequently asked
questions (FAQs) could also be shortened into 60-second short-form videos and released every week to stagger content release
and maintain audience engagement. By collaborating with the Communication and Outreach team, each Multi-Hazard and
Resilience department can contribute to creating their own FAQ series. Each FAQ session could feature five questions
(resulting in a total of 30 60-second videos if every Multi-Hazard and Resilience team decides to participate). Additionally,
these FAQs would serve as a valuable resource to journalists.

### 5.1.4.3 Video Titles

Video titles need to accurately describe the content of the video; however, titles play an important role in whether a video will
be clicked on and watched (Li et al., 2022; Cui et al., 2024). Some of the most popular videos released on the BGS YouTube
Channel are videos with titles referencing a recent geological event such as a volcanic eruption or landslide, questions/topics
that users may be searching for e.g., 'How rivers work: The role of groundwater' [Released: 22[nd] November 2013], 'Carbon
Capture and Storage research at the British Geological Survey' [Released: 17[th] November 2022] or may have enticing titles
that encourage users to want to know more e.g., 'Firing lasers into space 2' [Released: 3[rd] February 2017].

### 5.2.5 Public Feedback

BGS has upheld a valuable line of communication with the public since its inception and has maintained this by addressing
feedback/comments left on social media posts. However, it is noted that some questions/comments on social media posts
remain unanswered, this could be due to the nature of the Comment, questions falling outside of the Communication and
Outreach teams' expertise, the volume of Comments, and Comments being left on social media posts out of working hours.
To not fall into the issues identified by Su et al. (2017), it is suggested that engagement with the public is maintained.

### 5.3 Limitations

Regarding social media analysis, the lead author is the only researcher involved in creating and applying the codebook to BGS
social media posts. In previous studies, to maintain accuracy, multiple researchers have facilitated the analysis of online content
(Vu et al., 2021; Connoway et al., 2022). This allows researchers to compare categorisation and ensure accuracy in results.
This was not possible in this study, and thus, the chance for discrepancies between what the author and reader may consider a
post relating to 'Geodesy and Earth Observation'.



A small number of BGS employees took part in the semi-structured interviews due to time constraints and a requirement that the employee must be involved in voluntarily running a social media account associated with BGS. Employees that do not have a relationship with social media were not interviewed – this exclusion may result in the authors not hearing from diverse perspectives. Many factors need to be considered when voluntarily running social media accounts, including work-life balance,

voluntary roles typically falling on female employees (Babcock et al., 2017), and work priorities. These factors (plus many more) were unfortunately not captured during the interview process.

We acknowledge that sentiment and thematic analysis offer only a partial view of the dialogic potential of social media platforms, and that two-way communication requires both organisational responsiveness and integration of audience contributions into future practice. Our study seeks to understand patterns of audience behaviour and sentiment within the

constraints of publicly available data. We believe this still offers valuable insights for those seeking to enhance dialogic practice, even if our data do not allow us to evaluate internal response mechanisms or the degree to which BGS actions were shaped by audience input.

**5.4 Future Work**

To better understand BGS' social media communication, future work should focus on all of its social media activity – including discussion on themes not covered in this study such as decarbonisation, environmental geoscience, digital geoscience, accessibility, and internal communication. Understanding the intersections between these topics and the Multi-Hazard and Resilience campaigns will enlighten the overall BGS communication strategy.

To further investigate the dialogic potential of BGS's social media work, we propose a study of dialogic approaches to

communication by incorporating an analysis of organisational responses, which would significantly enhance our understanding of the dialogic model.

Although this study focusses on BGS – its research methodology and mixed-method approach can be applied to scientific organisations, institutions and individuals all around the world that want to analyse their current social media communication strategy and find ways for improvement.


**6 Conclusion**

This paper showcases the British Geological Survey's (BGS) successful establishment of an online presence across multiple social media platforms, accompanied by the creation of diverse and engaging content formats. Multi-Hazards and Resilience emerges as a captivating scientific topic that receives significant interest and curiosity amongst BGS' social media audience,

an interest that BGS effectively taps into throughout its numerous social media campaigns.



To answer RQ1; **What patterns of one-way engagement (likes, shares, comment volume) do BGS social media posts receive across different multi-hazard topics**? We carried out content and sentiment analysis on social media posts from May 2023 – March 2024 on X, LinkedIn, Facebook, Instagram, BlueSky and YouTube. Our results show a consistent output of Multi-Hazard and Resilience content across platforms (although a discrepancy is evident between short-form text and long-
form/video-based platforms). Our results also indicate that different Multi-Hazard and Resilience themes generate very different responses from the audience.

To answer RQ2; What challenges does BGS face in using social media to promote their research, and how can these be addressed to improve public engagement? We carried out semi-structured interviews with five BGS employees which self-identified as being involved in voluntarily running BGS-adjacent social media accounts. From the interviews, several
suggestions and ideas where discussed including improving lines of communication and increasing video output.

 Social media analysis and semi-structured interviews have been combined to paint a picture of a complex research environment, composed of several different themes, including the 'Multi-Hazard and Resilience' umbrella which encompasses eight areas of research.

Numerous suggestions have been proposed to foster increased social media engagement relating to Multi-Hazards and
Resilience and to enhance overall social media audience levels. These recommendations aim to further capitalise on the existing momentum and interest surrounding Multi-Hazard and Resilience, thereby amplifying BGS' impact and outreach in the realm of geoscience communication.

**Data Availability** The social media analytics and code book are available as an excel file on dair.dias.ie (a FAIR-aligned data repository). The file can be found at: https://dair.dias.ie/id/eprint/1463.

**Author Contribution** Eleanor Alice Dunn designed the research questions and carried out data analysis. Sam Illingworth assisted with the conceptual design and analysis. Jon-Paul Orsi assisted with data accessibility and analysis. Eleanor Alice Dunn prepared the manuscript with contributions from all co-authors.

**Competing Interests** The authors declare that there are no competing financial interests. However, it should be noted that one of the authors is the chief executive editor of Geoscience Communication.

**Ethical Statement** Ethical approval for this research was given by the British Geological Survey ethics committee.

**Acknowledgements** The authors would like to express their gratitude to all employees at the British Geological Survey who were interviewed as well as any employees who assisted with providing social media data.

**Financial Support** This research formed part of SPIN. SPIN is a European project funded by the European Union's Horizon 2020 research and innovation programme under the Marie Skłodowska-Curie grant agreement № 955515. The project is a
Marie Skłodowska-Curie Innovative Training Network (MSCA ITN): a joint research training and doctoral programme.



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
