# Peer review of "Leveraging Social Media for Geoscience Communication: Insights from the British Geological Survey's Multi-Hazard and Resilience Campaigns"

_EGUsphere, 2025_

## Referee Comment (RC2)

Title: Leveraging Social Media for Geoscience Communication: Insights from the British Geological Survey's Multi-Hazard and Resilience Campaigns

Author(s): Eleanor Alice Dunn et al.

MS No.: egusphere-2025-1963

MS type: Research article

**Review**

This paper explores how social media can be used in geoscience communication, particularly asking how effectively the British Geological Survey engage the public on multi-hazard / resilience focused work, and the challenges that BGS staff encounter when using social media platforms to enhance public understanding of this theme. The paper draws on existing methodologies to deliver these research objectives, combining analysis of social media post engagement with (a small number of) semi-structured interviews. Below are a set of major comments that I believe would significantly strengthen the paper, and minor comments that would improve overall readability and clarity.

**A. Major Comments**

**1. Independence (e.g., Line 59-62).** It feels odd that the paper suggests that it is an independent perspective on BGS' social media output and gives an objective overview of social media activity and its effectiveness because "the lead author is not a BGS employee" when another author (involved in the analysis and authorship, as set out in the author contributions) is a BGS employee. The independence of a paper is not solely shaped by the lead author's distance from the organisation it seeks to study. While not suggesting the authors (even if having connections with BGS) are unable to look objectively at the data – I would challenge the idea that the paper gives 'an independent perspective' and would strongly encourage more transparency about the measures taken to ensure objectivity and any implications of positionality of authors that may shape the recommendations and conclusions being made.

**2. Structure / Superfluous Paragraphs (e.g., Section 2.1)**

Overall, the paper would benefit from a careful proof-read and edit, to improve the flow of the narrative. There are some paragraphs that feel superfluous to the core focus of the paper (e.g., Section 2.1 - Why is this needed, giving a history of the journal you are seeking to publish in?), and some of the general background on X ownership and features (e.g., Lines 134-141). There may be other places where you can tighten up the flow and content also.

**3. Contribution beyond BGS (and engagement with wider literature)**

There are some sections where more engagement with the published literature would help the reader to understand how this work builds on others, what it is adding in terms of originality, and an evidenced explanation of how this work is of relevance beyond BGS. For example, you make very brief reference to your work being useful beyond BGS – but it's not clear how (i.e., what you have added to the methods and approaches set out in the wider literature).

**4. Data and Rationale for Assertions / Conclusions (e.g., Line 428-430)**

Various recommendations and assertions are made, but the data/evidence to support these is not always set out. Some specific examples are given below, but I think the article would benefit from a careful check throughout to ensure there is a clear link between each recommendation and the data (results), reflecting on the broader literature where relevant.

**5. Need for more nuanced reflections**

Different social media platforms have different audiences, and the article is lacking any significant engagement with what typical audiences may be and how that may impact on the way that posts are engaged with on different social media. For example, there may be substantially more engagement with BGS posts on Linked In by the wider professional geoscience community than on Facebook. I don't know if this is the case or if data exists to explore this – but I was surprised there wasn't more nuanced reflection on audiences in Sections 2, 4, and 5.

**B. Minor Comments / Suggestions**

**Line 58 (and elsewhere).** Here you note that the accounts are run on a 'voluntary' basis. It would be helpful to know what this means in practice, and have it defined. It may not be explicitly in their job description, but is something vaguer about communicating their science to diverse audiences described (for example)? Do people note this as an example of their communication / operational support in promotion applications / annual reviews, in which case it goes beyond a voluntary task?

**Line 79-80.** You note that the interviews (five) are used "to understand what barriers individuals may may face when using social media to promote their research and how these barriers can be tackled". This currently reads as if the perspectives of five individuals with tightly constrained selection criteria are being used to address a much broader question about barriers to promoting research. While those five individuals will have valid perspectives, it feels like that question needs to be explored with a bigger and more diverse audience to harness informative and more generally applicable recommendations.

**Line 106.** Insert the word 'about' between communication and research.

**Line 107-108.** What is the evidence for the claim in this sentence (that there is an increasing number of scientists who want to facilitate a valuable exchange....)?

**Line 112-116.** A claim is made about governmental organisations - this is then supported with papers from ~ a decade ago. Are there any more recent studies that explore whether this is still an issue. If not, can you add critical reflection on whether these sources still mirror the reality of 2025.

**Line 157-158.** I suggest adjusting the language from 'Facebook is... leading the way' to 'Facebook is host to substantive amounts of fake news...'. The former implies an intended objective, and I don't think this is substantiated in your manuscript.

**Line 200-204.** In your current structure, this sits under Section 2.3.5, but these lines are not specific to that section. Delete or merge with Section 3.

**Table 1.** Make clear in the caption that the accounts run by BGS Comms are just for the Multi-Hazard and Resilience Theme.

**Line 234.** Is a laughing emoji or reaction always positive? Are there not situations where one could be laughing out of sarcasm or ridicule, or because they found something bizarre?

**Figure 1.** This figure includes an acronym (LANDSLIP) of a very specific project which jars with the rest of the figure, which is more generic. Perhaps note in the caption that the arrows and annotations are exemplars (i.e., this is not a comprehensive overview of links).

**Line 273.** Please include the ethical review reference number / code (if given) and clarify in the text that ethical approval was obtained prior to commencing the work.

**Table 3.** For question 7, here and throughout, some nuanced discussion about what is and isn't in the job description may be helpful. See the earlier comment on **Line 58.**

**Figure 2.** The resolution could be improved, legend made larger, lines made less feint (very hard to see for the yellows and oranges, particularly).

**Line 300.** I'm not sure if bias is an appropriate word here, it implies a deliberate or inadvertent favouring – but there may be deliberate and fair reasons why certain departments are better represented over the relatively short time frame of the study.

**Line 303.** I don't think 'popular area' is an appropriate phrase here – can you think again about this sentence.

**Line 304.** Change "% on" to "% of posts on".

**Figure 4.** It is not easy for a reader to engage with this Figure. The legend is too small, the numbers on chart itself are too small and difficult to read.

**Figure 5.** It is not easy for a reader to engage with this Figure. The blue / teal colours are all very similar. In lines 309-310 you note that the likes are normalised, but it's not mentioned in the figure caption (nor clear if and how this has been done).

**Line 337-338.** Here a recommendation is made to increase posts, and to streamline content best suited to different platforms. This seems out of place in the results, not well evidenced (how did you get from result to recommendation) and again, what role does audience have on the nature of engagement?

**Line 344.** Remove word 'must'.

**Table 5.** What is NSFW?

**Line 374.** Unclear if you mean the interviewee's enjoyment managing the account or the enjoyment of users following the account?

**Line 389.** Again, do you think the suggestions should be dotted in the results and discussion, or grouped together in the latter? With this example, please add a little more detail… "it is suggested [by who?] that support [what support?] be provided… will enable the continued success of the account" [what is the evidence for ongoing success – given you use the word continued – and that the suggestion will enable this?].

**Line 396.** Change 'many' to 'may'

**Line 402-403.** I'm not sure this 'clear interest' is evidenced by the previous bullet points (it seems to contrast with bullet #2, Line 399).

**Section 5.1.1.** (a) It's not clear why this is being posed as a problem and needing corrective action. (b) On the recommendation of having an 'on call' roll - there is huge diversity in the group described, and it is not clear how a landslide expert could offer advice/support on questions/comments regarding geomagnetism, for example.

**Line 423.** I'm unclear what 'underutilised market' in this context means – a market for what purpose?

**Line 427.** Change 'regarding' to 'compared to' (or something similar).

**Line 428-430.** What is the evidence for this assertion (re. BlueSky)?

**Line 429.** Change 'consisting' to 'consistent'.

**Line 436.** What is the evidence for this assertion (re. videos)?

**Line 449.** Check use of word 'present' – this doesn't read well.

**Line 485.** How many questions/comments were unanswered? Out of how many? The evidence for including this suggestion could be strengthened.

**Line 494-495.** Update to 'and thus, there is a chance for discrepancies to exist between…' and add (for example) at the end of the sentence.

**Lines 505-507.** This sentence is not clear and may need to be rewritten. Can you also clarify if the beneficiaries of these insights are those within or beyond BGS, and if beyond the evidence for that / limitations.

**Line 517.** This sentence seems to imply that you are presenting a methodology that is new – and can be applied elsewhere. Have these approaches not been applied before? Can you be explicit in this section about what you are offering that is novel in terms of methodology, vs what existing methods you are applying.

**Line 535.** Change 'where' to 'were'.

**Ethical Statement.** Can you add in the ethical approval reference number if you have it.

**Checks throughout:**

- Both X (formerly Twitter) and just Twitter are used – please be consistent.
- Capitalisation of some terms (Likes, Shares, Comments) is not aways consistently applied.
- This is a multi-authored piece, so use of 'I' needs to be removed (e.g., **line 362**, maybe elsewhere).

- The piece needs an additional proof-read prior to resubmission, with some incorrect phrasing / missing words used (examples given above).